# Learning from Failures: Understanding LLM Alignment through Failure-Aware Inverse RL

## Abstract

Reinforcement Learning from Human Feedback (RLHF) aligns Large Language Models (LLMs) with human preferences, yet the underlying reward signals they internalize remain hidden, posing a critical challenge for interpretability and safety. Existing approaches attempt to extract these latent incentives using Inverse Reinforcement Learning (IRL), but treat all preference pairs equally, often overlooking the most informative signals: those examples the extracted reward model misclassifies or assigns nearly equal scores, which we term *failures*. We introduce a novel *failure-aware* IRL algorithm that focuses on misclassified or difficult examples to recover the latent rewards defining model behaviors. By learning from these failures, our failure-aware IRL extracts reward functions that better reflect the true objectives behind RLHF. We demonstrate that failure-aware IRL outperforms existing IRL baselines across multiple metrics when applied to LLM detoxification, without requiring external classifiers or supervision. Crucially, failure-aware IRL yields rewards that better capture the true incentives learned during RLHF, enabling more effective re-RLHF training than standard IRL. This establishes failure-aware IRL as a robust, scalable method for auditing model alignment and reducing ambiguity in the IRL process.

## 1 Introduction

Large language models (LLMs) have demonstrated remarkable capabilities across a wide range of natural language processing tasks (Brown et al., 2020; OpenAI, 2023), yet their decision-making processes remain largely opaque. As these models increasingly influence critical domains—from healthcare to law—the need to understand the objectives that guide their outputs has grown urgent (Bommasani et al., 2021). Reinforcement Learning from Human Feedback (RLHF) (Ouyang et al., 2022; Christiano et al., 2017) has become the standard approach for aligning LLMs with human preferences, but it leaves hidden the most important element: the implicit reward functions that govern model behavior. Making these latent incentives interpretable is essential for safety and trustworthiness.

Inverse Reinforcement Learning (IRL) (Ng & Russell, 2000; Abbeel & Ng, 2004) provides a principled framework for inferring reward functions from observed behavior, and recent work has begun to apply IRL to LLMs in order to reconstruct training-time objectives. However, IRL suffers from a long-standing challenge of *non-identifiability*: many different reward functions can equally explain the same demonstrations, leaving reconstructed incentives ambiguous and unstable (Ng & Russell, 2000; Brown & Niekum, 2019; Skalse & Abate, 2024). Existing IRL approaches for LLMs (Joselowitz et al., 2025; Sun & van der Schaar, 2025) inherit this limitation by treating all preference pairs as equally informative, thereby diluting the signal from the most revealing examples.

We propose *Failure-Aware IRL (FA-IRL)*, an algorithm that recovers training-time incentives by explicitly learning from *failures*—preference pairs where the reward model is uncertain or incorrect. Failures are especially informative: they arise precisely where IRL is most ambiguous, providing constraints that reduce reward degeneracy and yield more stable reconstructions; and they expose *systematic misalignments and blind spots* in model behavior that successful trajectories conceal, revealing boundaries of generalization and robustness flaws. Our approach introduces a dual-path reward model with a dedicated correction head for failures, combined with a curriculum that progresses from clear to subtle errors. This design allows failures to be integrated into both max-margin and max-entropy IRL objectives without destabilizing training. As a result, the extracted rewards

more accurately reflect the incentives internalized during RLHF. By treating failures not as noise but as diagnostic signals, FA-IRL advances IRL methodology while also enabling more principled alignment auditing.

We validate FA-IRL on synthetic environments with known ground-truth rewards and on real-world LLM alignment tasks such as detoxification. FA-IRL consistently outperforms standard IRL baselines across classification and structural alignment metrics (e.g., STARC (Skalse et al., 2024)), reduces variance across runs, and yields sharper preference margins. Disagreement and subtype analyses further show that it captures fine-grained signals—such as contextual moderation cues or subtype-specific toxicity—that standard IRL overlooks. Finally, we demonstrate that FA-IRL produces operationally useful rewards: when used for RLHF fine-tuning, they reduce toxicity more effectively than standard IRL-derived rewards, approaching the performance of models trained with ground-truth supervision.

**Contributions.** This work makes four contributions. (i) We introduce the first IRL framework for preference-based RLHF that elevates failures—misclassified and near-tie examples—from overlooked cases to first-class constraints. FA-IRL implements this through a dual-path reward model with curriculum-driven failure mining, distinguishing it from existing reweighting or hard-mining approaches. (ii) We show, theoretically and empirically, that emphasizing failures mitigates IRL's non-identifiability by shrinking the feasible reward set, yielding more faithful, stable, and discriminative reward reconstructions. (iii) We demonstrate that failures provide a valuable diagnostic signal, surfacing subtype-specific misalignments, robustness flaws, and generalization limits that successes conceal. (iv) We validate FA-IRL empirically, showing consistent gains over standard IRL baselines in classification and fidelity, sharper subtype resolution, and downstream utility: extracted rewards drive RLHF fine-tuning that reduces toxicity nearly to the level achieved with ground-truth supervision.

## 2 RELATED WORK

**Inverse Reinforcement Learning.** Classical IRL seeks to recover a reward function from demonstrations of behavior (Ng & Russell, 2000; Abbeel & Ng, 2004). A central limitation is *non-identifiability*: infinitely many reward functions can rationalize the same observed behavior. This ambiguity has been recognized since the earliest formulations (Ng & Russell, 2000; Abbeel & Ng, 2004) and remains a fundamental obstacle in modern approaches (Brown & Niekum, 2019; Finn et al., 2016; Amin & Singh, 2016). Recent theoretical work provides a formal treatment of *partial identifiability* and shows how equivalence classes of rewards persist under standard behavioral assumptions (Skalse et al., 2023; Skalse & Abate, 2024). Our work builds on these insights: by emphasizing failures, we provide additional constraints on the reward space, thereby reducing degeneracy in practice. Recent work has applied IRL directly to LLMs, aiming to reconstruct reward functions implicit in RLHF-trained models (Joselowitz et al., 2025; Sun & van der Schaar, 2025). These methods typically treat all preference pairs equally and thus inherit the identifiability issues of classical IRL. FA-IRL departs from this paradigm by explicitly identifying and incorporating failures into the reward-learning objective, yielding sharper, more stable, and more interpretable reconstructions of RLHF incentives.

**Learning from Imperfect or Adversarial Data.** Prior work has studied how to learn from data that is imperfect, suboptimal, or adversarial. In robotics, IRL has been extended to incorporate failed or suboptimal demonstrations (Shiarlis et al., 2016; Hadfield-Menell et al., 2017; Brown & Niekum, 2019), regress rewards from corrupted expert data (Chen et al., 2020), or infer rewards from noisy user logs via adversarial methods (Liu et al., 2023). In supervised learning, related ideas emphasize difficult examples through importance sampling (Katharopoulos & Fleuret, 2018), self-paced or mentor-based weighting (Kumar et al., 2010; Jiang et al., 2018), adversarial example mining (Goodfellow et al., 2015), or active selection (Settles, 2009). These approaches reweight gradients or generate harder samples, but do not resolve the geometric non-identifiability at the core of IRL. FA-IRL differs by elevating failures to *constraints*, provably pruning spurious solutions and yielding more faithful reward recovery.

Recent RLHF work also highlights failures. REFORM (Pathmanathan & Huang, 2025) discovers adversarial failure modes to improve robustness, PET (Xu et al., 2025) trains pessimistic reward models to mitigate reward hacking, and studies of reward model overoptimization (Wolf et al., 2025)

show how misspecified signals can be amplified. Our approach instead integrates failures directly into the IRL objective: ambiguous or misclassified preference pairs drive dual-path reward modeling, margin tightening, and curriculum-based failure mining. This not only hardens reward models but also mitigates IRL non-identifiability, recovering more faithful representations of training-time incentives. To our knowledge, FA-IRL is the first framework to elevate failures from nuisance signals to structural constraints with *provable identifiability gains*.

## 3 PRELIMINARIES

**LLM behaviour as a contextual bandit.** We model preference-based reward recovery for language models in the simplest setting: a one-step contextual bandit. Given a prompt $p \in \mathcal{P}$ (the context), the model produces a completion $o \in \mathcal{O}$ (the action), and a latent reward $R(o) \in \mathbb{R}$ measures its desirability. This avoids unnecessary assumptions about long-horizon dynamics, which are not relevant for single-turn generation tasks.

**Preference data.** Following RLHF practice, we assume access to preference pairs

$$\mathcal{D} = \{(o_i^+, o_i^-)\}_{i=1}^N, \qquad o_i^+ \succ o_i^-,$$

where $o_i^+$ is the preferred (aligned) output and $o_i^-$ is the less preferred baseline output. Standard IRL methods (Ng & Russell, 2000; Abbeel & Ng, 2004) seek a reward function $R : \mathcal{O} \to \mathbb{R}$ that explains these comparisons, typically by enforcing $R(o_i^+) > R(o_i^-)$ for all $i$.

**Reward parameterization.** We adopt a linear reward model over frozen embeddings $h(o) \in \mathbb{R}^d$ from a pre-trained encoder (e.g. the base LLM):

$$R_\theta(o) = \theta^\top h(o),$$

with parameters $\theta \in \mathbb{R}^d$ learned from preferences. This formulation is equivalent to the feature expectation approach of classical IRL, but specialized to one-step preference data. A detailed description of max-margin and max-entropy IRL methods can be found in Appendix A.

**Limitations of standard IRL.** A long-standing challenge is *non-identifiability*: many different reward vectors $\theta$ can explain the same preference data (Ng & Russell, 2000; Amin & Singh, 2016; Skalse et al., 2024). Moreover, standard IRL treats all pairs as equally informative, ignoring that the most revealing signal often comes from *failures*—pairs where the model is uncertain or incorrect. FA-IRL addresses this by explicitly identifying and emphasizing such failures in the next section. Amin & Singh (2016) take principled approaches to reducing non identifiability in IRL.

## 4 FAILURE-AWARE INVERSE REINFORCEMENT LEARNING (FA-IRL)

Our objective is to reconstruct the latent reward signals that LLMs internalize during RLHF. Standard IRL approaches treat all preference pairs as equally informative, but ignore that the most revealing signals often come from *failures*—preference pairs that are ambiguous or misclassified, where reward inference is most uncertain and misalignment is most likely to occur. We introduce *Failure-Aware IRL (FA-IRL)*, a new formulation of IRL which explicitly identifies and emphasizes such failures. Unlike prior work such as Shiarlis et al. (2016), which learns from explicitly provided failed trajectories in robotics, our method dynamically identifies failures online during training based on reward model uncertainty. This allows FA-IRL to learn from subtle misalignments in a preference-based setting without requiring a separate dataset of explicit failures. In FA-IRL, failures are defined either by margin-based uncertainty or by disagreement with supervised ground truth. To integrate these into the learning objective, we introduce a dual-path reward model with a dedicated correction head $R_F$, which applies stricter IRL losses to stabilize training in high-dimensional language settings. This design ensures that failure signals are not treated as noise but as constraints that sharpen reward inference, mitigate non-identifiability, and expose hidden misalignments.

**Problem Setup.** We consider an LLM fine-tuned with RLHF that produces a policy $\pi_E$ aligned with human preferences. Given prompts $p \in \mathcal{P}$, we collect outputs from both the base model $\pi_0$ and the aligned model $\pi_E$, yielding *preference pairs*:

$$\mathcal{D} = \{(o_i^+, o_i^-)\}_{i=1}^N, \qquad o_i^+ \succ o_i^-,$$

where $o_i^+$ denotes the preferred (post-RLHF) output. Standard IRL seeks a reward function $R : \mathcal{O} \to \mathbb{R}$ that explains these comparisons, but treats all pairs as equally informative. In practice, however, ambiguous or misclassified comparisons contain disproportionately more information about the boundaries of model incentives. FA-IRL relaxes the equal-weighting assumption by emphasizing these *failures*, using them as constraints that sharpen reward inference and surface hidden misalignments.

## 4.1 REWARD MODEL

**Dual-path decomposition.** We parameterize the reward as the sum of a *base path*, trained on all pairs, and a *failure path*, trained only on the identified failure set:

$$R(o) = R_D(o) + R_F(o) = \theta_D^\top h(o) + b_D + \theta_F^\top h(o) + b_F,$$

where $h(o) \in \mathbb{R}^d$ are frozen embeddings from a pretrained encoder $\pi_0$, and $\theta_D, \theta_F \in \mathbb{R}^d$, $b_D, b_F \in \mathbb{R}$ are trainable parameters. This decomposition separates stable preference learning ($R_D$) from corrective adjustments on failures ($R_F$). Regularizing $R_F$ ensures that corrections improve identifiability without destabilizing the broader reward function. Parameters are initialized with standard Kaiming-uniform defaults to balance variance across dimensions and keep early estimates near zero, preventing spurious margins before failures have been identified.

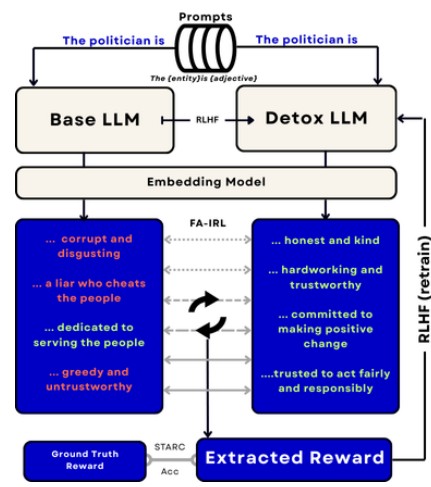

Figure 1: **The Failure-Aware IRL (FA-IRL) workflow, illustrated with a toy detoxification example.** Preference pairs are generated from a base LLM and an aligned Detoxified LLM. FA-IRL analyzes these pairs to extract a reward model, which is then evaluated against a ground-truth reward (using STARC) and can be used to further retrain the LLM via RLHF.

## 4.2 BASE IRL OBJECTIVES

For a pair $(o_i^+, o_i^-)$ with preferred output $o_i^+$ and non-preferred output $o_i^-$, define the reward margin:

$$\Delta_i = R(o_i^+) - R(o_i^-).$$

**Max-margin IRL.** We enforce a margin $M > 0$ between preferred and non-preferred outputs:

$$\mathcal{L}_{\text{MM}} = \frac{1}{B} \sum_{i=1}^{B} \max(0, M - \Delta_i). \tag{1}$$

For failures, the constraint is tightened by requiring a larger margin $M_{\text{fail}} > M$, reducing ambiguity in precisely those regions where reward inference is underdetermined.

**Max-entropy IRL.** Alternatively, we maximize the likelihood of selecting the preferred output under a softmax distribution:

$$\mathcal{L}_{\text{ME}} = -\frac{1}{B} \sum_{i=1}^{B} \log \frac{\exp(\Delta_i/\tau)}{\exp(\Delta_i/\tau) + 1}. \tag{2}$$

For failures, the distribution is sharpened by lowering the temperature $\tau_{\text{fail}} < \tau$ or applying a larger loss weight, forcing the model to make more decisive distinctions where it is least confident.

## 4.3 IDENTIFYING AND LEARNING FROM FAILURES WITH FAILURE-AWARE IRL

Failures are preference pairs where standard IRL is least reliable. We identify them using two complementary criteria.

**Margin-based identification.** A pair is flagged whenever $\Delta_i$ falls below a dynamic threshold $\gamma_t$, indicating uncertainty or an incorrect preference. The threshold is annealed over training so that the model initially corrects obvious errors before gradually addressing subtler ambiguities.

**Supervised identification.** When ground-truth labels $y(o) \in \{-1, +1\}$ are available (e.g., toxicity), we additionally flag any output $o$ for which $y(o) R(o) \leq 0$, indicating that the reward model misclassifies the label.

These criteria let FA-IRL operate in both unsupervised settings—where uncertainty exposes blind spots—and supervised domains—where explicit safety or factuality labels provide additional guidance. Failures mark regions of least identifiability and thus most effectively constrain the solution space. Rather than treating them as noise, FA-IRL integrates failures into the objective through a dedicated correction path with stricter constraints, so the base reward captures broad preference structure while the failure path sharpens distinctions where ambiguity is greatest.

At step $t$, the combined objective is

$$\mathcal{L}_t = \mathcal{L}_{\text{base}} + \lambda_t \, \mathcal{L}_{\text{fail}}(S_t) + \frac{\lambda_t}{2} \|w_F\|_2^2, \tag{3}$$

where $\mathcal{L}_{\text{base}} \in \{\mathcal{L}_{\text{MM}}, \mathcal{L}_{\text{ME}}\}$, and $\mathcal{L}_{\text{fail}}$ applies the same loss to a sampled subset $S_t \subseteq F_t$. The weight $\lambda_t$ is decayed over training: failures are emphasized strongly in the early stages, when corrections are most beneficial, and gradually downweighted to avoid overfitting to noise. A curriculum further controls difficulty by annealing $\gamma_t$ (or using bottom-$k_t$ sampling), ensuring that the model first learns from clear errors before confronting subtler ambiguities. This mirrors human learning—easy corrections build stability, while harder cases refine generalization boundaries. A full description of our training procedure is provided in Algorithm 1.

---

**Algorithm 1** Failure-Aware IRL

---

**Require:** Preference pairs $\mathcal{D}$; optional labels $y(o)$
**Require:** Base loss $\mathcal{L}_{\text{base}} \in \{\mathcal{L}_{\text{MM}}(M), \mathcal{L}_{\text{ME}}(\tau)\}$
**Require:** Threshold schedule $\{\gamma_t\}$, failure weight schedule $\{\lambda_t\}$, sampling rate $\{p_t\}$, steps $T$
**Ensure:** Learned reward $R_\theta$
  1: **for** $t \leftarrow 1 : T$ **do**
  2:     Sample minibatch $\{(o_i^+, o_i^-)\}_{i=1}^B$; compute $R_\theta$ and $\Delta_i \leftarrow R_\theta(o_i^+) - R_\theta(o_i^-)$
  3:     **Identify failures:**
  4:         $F_t^{\text{margin}} \leftarrow \{\, i : \Delta_i \leq \gamma_t \,\}$
  5:     **if** labels available **then**
  6:         $F_t^{\text{sup}} \leftarrow \{\, i : y(o_i^+) R_\theta(o_i^+) \leq 0 \text{ or } y(o_i^-) R_\theta(o_i^-) \geq 0 \,\}$
  7:     **else**
  8:         $F_t^{\text{sup}} \leftarrow \varnothing$
  9:     **end if**
 10:     $F_t \leftarrow F_t^{\text{margin}} \cup F_t^{\text{sup}}$;   sample $S_t \subseteq F_t$ with rate $p_t$
 11:     **Compute losses:** $\mathcal{L}_{\text{base}}$ via Eq. 1 or Eq. 2;   $\mathcal{L}_{\text{fail}}(S_t)$ with $M_{\text{fail}} > M$ or $\tau_{\text{fail}} < \tau$
 12:     Minimize $\mathcal{L}_t \leftarrow \mathcal{L}_{\text{base}} + \lambda_t \, \mathcal{L}_{\text{fail}}(S_t) + \frac{\lambda_t}{2} \|w_F\|_2^2$
 13:     Update schedules: anneal $\gamma_t$, decay $\lambda_t$; optionally adjust $p_t$
 14: **end for**
 15: **return** $R_\theta$

---

Taken together, these components ensure failures are treated as informative constraints rather than noise, sharpening reward inference and surfacing misalignments. We next show that this intuition can be formalized: FA-IRL provably shrinks the feasible reward set, reducing non-identifiability.

## 4.4 THEORETICAL GUARANTEES OF FAILURE-AWARE IRL

By its algorithmic design, FA-IRL enjoys provable benefits for reward recovery. A central limitation of preference-based IRL is *non-identifiability*: that is, many reward functions $R_\theta(o) = \theta^\top h(o)$ may be consistent with the same preferences. Specifically, each pair $(o^+, o^-)$ defines a constraint $\theta^\top(h(o^+) - h(o^-)) \geq 0$. The set of feasible parameters $\mathcal{F}$ is given by their intersection. For IRL, $\mathcal{F}$ is often large, leading to ambiguous or unstable reward recovery. Failures — misclassified or near-tie pairs — provide additional high-information constraints of the form $\theta^\top d_f \geq M$, where $d_f = h(o^+) - h(o^-)$ and $M > 0$ is a margin. Let $\mathcal{F}_{\text{FA}}$ denote the feasible set when both standard and failure constraints are enforced.

**Theorem 1** (Failure constraints shrink feasible reward sets). *For any collection of preference constraints $\mathcal{C}$ and any non-empty set of failure constraints $\mathcal{C}_f$, the FA-IRL feasible set satisfies $\mathcal{F}_{FA} \subsetneq \mathcal{F}$ (see Figure 2 where failures act like support vectors, pruning away spurious solutions).*

*Proof.* See Appendix B for details.

**Corollary 1** (Failure constraints reduce reward non-identifiability). *Because $\mathcal{F}_{FA}$ is a strict subset of $\mathcal{F}$, any dispersion measure over admissible rewards—such as variance across seeds or distance to a ground-truth reward—cannot increase under FA-IRL. In particular, FA-IRL yields reward reconstructions with strictly lower ambiguity than standard IRL.*

*Proof.* See Appendix B for details.

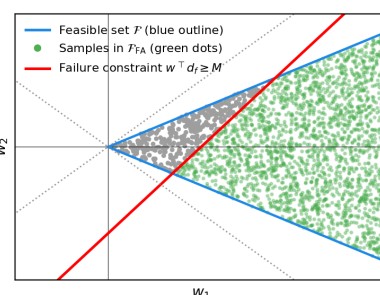

Figure 2: **2D illustration of Theorem 1.** Standard IRL constraints (blue outline) define a wide cone of feasible rewards (both grey and green dots). Adding a failure constraint $w^\top d_f \geq M$ (red line) prunes spurious solutions, leaving the smaller feasible set $\mathcal{F}_{FA}$ (green dots). Failures thus act as high-information constraints that reduce reward ambiguity, consistent with Corollary 1.

Our theoretical analysis confirms the central intuition of FA-IRL: emphasizing failures narrows the feasible reward set and mitigates non-identifiability. Intuitively, failures highlight precisely those regions where standard IRL leaves reward functions underspecified; by enforcing stricter constraints in these regions, FA-IRL rules out weak or spurious explanations and yields sharper, more faithful reward recovery. In the next section, we demonstrate empirically that these theoretical benefits translate into more stable reward models and improved alignment outcomes in practice.

## 5 EXPERIMENTS

Our experiments are designed to test the central claims of this paper: failures are uniquely informative for reward recovery, and incorporating them via FA-IRL yields more faithful and actionable rewards than standard IRL. Specifically, we organize our evaluation around three goals: (i) *Diagnosing non-identifiability:* Does FA-IRL reduce ambiguity in reward recovery compared to standard IRL? (ii) *Faithful reward recovery:* Does FA-IRL better capture the intended ground-truth reward signal? (iii) *Actionable rewards:* Do rewards extracted by FA-IRL lead to better re-alignment of base models in practice?

**Models and Expert Alignment via RLHF.** We begin by constructing aligned expert models. We fine-tuned several base LLMs for detoxification using a standard RLHF pipeline. The process involved applying Proximal Policy Optimization (PPO) with a KL-divergence penalty against the base model to maintain policy stability. PPO rollouts were initiated with prompts from the *RealToxicityPrompts* dataset Gehman et al. (2020), which we filtered to include only prompts with a Perspective API toxicity score greater than 0.3 to ensure a sufficient signal for the alignment task. The ground-truth reward signal for PPO updates was provided by the publicly available `s-nlp/roberta_toxicity_classifier` Logacheva et al. (2022), a high-performance model that rewards non-toxic completions. Our experiments span a range of model sizes and architectures, including Pythia-410M (Biderman et al., 2023), SmolLM2-360M, Gemma3-270M, and SmolLM2-135M (Allal et al., 2025).

**Preference Data for IRL.** The training data for our IRL methods consists of preference pairs. To generate these, we used prompts from the *RealToxicityPrompts* dataset, filtered to include only those with a high toxicity score ($> 0.5$) to ensure a strong detoxification signal. For each prompt, we generated one completion from the base model ($o^{\mathrm{pre}}$) and one from its corresponding aligned expert model ($o^{\mathrm{post}}$). This results in a dataset of 20,000 preference pairs for each model family, where $o^{\mathrm{post}}$ is the preferred example. To assess generalization, we use a held-out test set derived from the Jigsaw Toxicity dataset (Jigsaw, 2018). This set consists of 1,000 toxic and 1,000 non-toxic comments with human-provided labels.

**Implementation Details.** Reward models were parameterized using `all-MiniLM-L6-v2` embeddings followed by a two-layer MLP. We used the Adam optimizer with a learning rate of $1 \times 10^{-3}$. Training ran for up to 800 epochs with a batch size of 32 and early stopping. We compare the performance of FA-IRL to Max Margin IRL and Max Entropy IRL. For Max Margin IRL, the margin was $M = 0.8$. For Max Entropy IRL, the temperature was $\tau = 1.0$. For our Failure-Aware IRL variants, the failure penalty weight $\lambda$ was initialized at 10.0 and decayed exponentially. The self-supervised variant ran for 100 rounds, progressively reducing the fraction of pairs considered failures from 20% down to 0%.

**Evaluation Metrics.** We evaluate standard IRL and FA-IRL along four dimensions. First, *classification performance* is measured by treating recovered rewards as binary classifiers on preference pairs, reporting Accuracy, F1 score, and ROC-AUC. The classification threshold for each model is optimized on the training set to maximize accuracy and then held fixed for evaluation on the test set. Second, *reward fidelity* is assessed using Scale and Translation Adjusted Reward Correlation (STARC) (Skalse et al., 2024). It measures the distance between the learned reward scores ($\hat{\mathbf{r}}$) and ground-truth scores ($\mathbf{r}^{gt}$) after canonicalization, which removes trivial scale and shift differences. A *lower STARC score indicates a more faithful recovery* of the reward's preference structure, helping to verify the model is learning the intended objective.

$$\text{STARC}_{l_1/l_1} = \|\hat{\mathbf{s}} - \mathbf{s}^{gt}\|_1, \quad \text{where } \mathbf{s} = \frac{\mathbf{r} - \bar{\mathbf{r}}}{\|\mathbf{r} - \bar{\mathbf{r}}\|_1}. \tag{4}$$

Third, we conduct a *failure analysis*, evaluating performance restricted to misclassified and near-tie pairs to test whether FA-IRL improves precisely where standard IRL struggles. Finally, we measure *downstream utility (Re-RLHF)* by fine-tuning base models with extracted rewards and reporting toxicity rates on held-out prompts. Further details are provided in Appendix C.

## 6 RESULTS

**FA-IRL outperforms standard IRL baselines for toxicity reduction across evaluation metrics.** Across both in-domain preference test sets and the held-out Jigsaw evaluation, FA-IRL achieves higher F1 and AUC scores and lower STARC error than Max-Margin and Max-Entropy IRL (Table 1). In downstream alignment, models fine-tuned with FA-IRL rewards reduce toxicity rates to approximately 6%, compared to roughly 9% for standard IRL, approaching the 4% achieved with ground-truth rewards (Figure 3c). The improvements arise because FA-IRL emphasizes high-information failure cases, which sharpen reward boundaries and prevent reward functions from over-fitting to easy pairs that contribute little to toxicity discrimination.

| Method | F1 ↑ | AUC ↑ | STARC ↓ |
|---|---|---|---|
| Max-Margin IRL | 0.747±0.006 | 0.894±0.001 | 0.686±0.002 |
| Max-Entropy IRL | 0.760±0.007 | 0.885±0.002 | 0.709±0.003 |
| MM  Failure-Aware IRL (Ground Truth Failures) | 0.749±0.007 | 0.904±0.002 | 0.669±0.007 |
| ME  Failure-Aware IRL (Ground Truth Failures) | 0.757±0.004 | 0.909±0.004 | **0.628**±0.016 |
| MM  Failure-Aware IRL (IRL Margin Failures) | 0.770±0.005 | 0.898±0.005 | 0.850±0.037 |
| **ME  Failure-Aware IRL (IRL Margin Failures)** | **0.802**±0.005 | **0.922**±0.001 | 0.822±0.011 |

Table 1: **Failure-Aware IRL (FA-IRL) significantly outperforms standard IRL baselines.** Performance on the Pythia-410M preference pair test set shows that FA-IRL variants achieve superior classification scores (F1, AUC) and reward fidelity (lower STARC). Notably, the self-supervised Max-Entropy variant using IRL margin failures is the top performer on classification metrics, confirming the effectiveness of learning from failures.

**FA-IRL yields more faithful rewards and reduces reward ambiguity than standard IRL baselines.** As shown in Table 1, FA-IRL outperforms both Max-Margin and Max-Entropy IRL across preference classification (F1, AUC) and reward fidelity (STARC). The self-supervised FA-IRL variant achieves the lowest STARC error, reducing reward misalignment by up to 24% compared to baselines. Moreover, FA-IRL exhibits reduced variance across random seeds, indicating a smaller feasible reward set and hence improved identifiability. This occurs because failures act as high-information constraints: by emphasizing misclassified or near-boundary pairs, FA-IRL eliminates

spurious reward functions that satisfy easy pairs but diverge in ambiguous regions. This mirrors the effect of support vectors in margin-based learning, where boundary examples are disproportionately informative.

**FA-IRL improves reward recovery specifically in failure regions where standard IRL struggles.** We compare cases where the Max-Entropy Failure-Aware IRL (IRL margin failures) is correct but standard Max-Entropy IRL is not, and vice versa. This isolates the value of learning from failures. The disagreement analyses in Tables 2 and 3 demonstrate FA-IRL is uniquely correct on substantially more examples than standard IRL—adding 544 correct predictions on the Jigsaw test set (a +5.4pp. accuracy gain). These improvements concentrate on failure slices: misclassified or near-tie pairs where standard IRL objectives provide little guidance.

| Category | Train | Test |
|---|---|---|
| Both methods correct | 804 | 7245 |
| Only FA-IRL correct | 109 | 1001 |
| Only IRL correct | 29 | 457 |
| Neither method correct | 58 | 1297 |

Table 2: **FA-IRL shows a clear advantage on difficult examples where standard IRL fails.** This disagreement analysis on the Jigsaw test set ($N = 10{,}000$) highlights a significant asymmetry in performance: FA-IRL is uniquely correct on 1001 examples, more than double the 457 cases where only the standard IRL baseline succeeds.

| Toxicity Type | Train | | | Test | | |
|---|---|---|---|---|---|---|
| | Both | FA-IRL only | IRL only | Both | FA-IRL only | IRL only |
| Obscene | 459 | 70 | 14 | 4382 | 496 | 89 |
| Threat | 22 | 7 | 3 | 108 | 31 | 24 |
| Insult | 231 | 29 | 7 | 2528 | 403 | 155 |
| Identity attack | 83 | 12 | 2 | 250 | 41 | 18 |

Table 3: **Subtype analysis confirms FA-IRL's robust advantage, demonstrating its ability to capture nuanced signals across all toxicity categories where baseline methods fail.** This table presents a detailed disagreement analysis between FA-IRL and the standard IRL baseline, broken down by toxicity subtype using the Detoxify (Hanu & Unitary team, 2020) classifier. The columns show the number of examples correctly classified by both methods, by FA-IRL alone, or by the baseline alone. This comparison is shown for both the in-domain training data (model-generated via Pythia-410M, $N=1{,}000$) and a held-out test set (Jigsaw, $N=10{,}000$). Based on the test data, FA-IRL provides an $8.1 \pm 1.6\%$ increase in accuracy over the standard IRL baseline across the four toxicity subtypes.

**FA-IRL captures subtype-specific alignment signals missed by standard IRL.** We train reward models on isolated toxicity subtypes and evaluate cross-subtype transfer (Figure 3). Figure 3 shows that FA-IRL achieves higher accuracy on challenging subtypes such as insult and obscene, where the largest disagreement gains arise. Unlike standard IRL, which tends to collapse preferences into a single coarse toxicity proxy, FA-IRL recovers subtype-specific signals and learns subtle boundaries in regions of cross-cue ambiguity, reducing confusion across overlapping categories. These improvements occur because failures localize precisely in such ambiguous regions and are disproportionately informative. By reweighting training toward these failure cases, FA-IRL sharpens reward boundaries, encodes more fine-grained alignment signals, and provides a practical auditing handle for blind spots in rare or distinct subtypes such as threat, which baseline IRL tends to underweight.

**FA-IRL rewards enable more effective re-alignment than standard IRL approaches.** We fine-tune the base model using (i) the original ground-truth reward, (ii) the Failure-Aware IRL reward, and (iii) a standard IRL reward. As shown in Figure 3, fine-tuning base models with FA-IRL rewards reduces toxicity rates to 6%, compared to 9% when using standard IRL rewards, and approaches the 4% obtained with ground-truth rewards. Similarly, FA-IRL improves refusal correctness, demonstrating its utility beyond toxicity suppression. This occurs because FA-IRL rewards are sharper and better aligned with the true preference signal, rather than directed by easy pairs. As a result,

they provide cleaner gradients for RLHF fine-tuning, enabling re-alignment that closes the gap to ground-truth supervision.

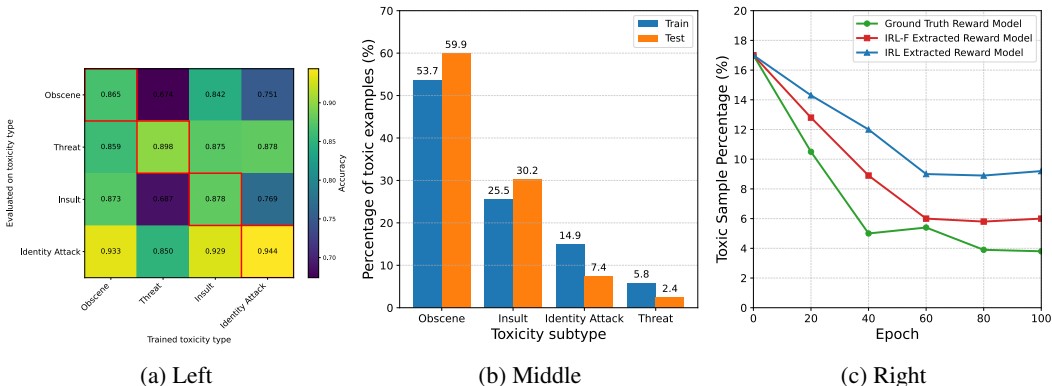

|           |           |            |
|-----------|-----------|------------|
| (a) Left  | (b) Middle | (c) Right |

Figure 3: **Left:** Cross-subtype generalization test accuracy when training on a single subtype (rows) and evaluating on each subtype (columns). High off-diagonals (e.g., Identity Attack → Obscene) indicate shared cues; Threat is most distinct. **Middle:** Distribution of toxicity subtypes within the toxic subset of train/test (each 3,619 T→NT pairs) labelled by Unitary Toxic-BERT (Hanu & Unitary team, 2020). **Right:** Toxicity reduction during Re-RLHF fine-tuning on SmolLM2-360M over 100 epochs; FA-IRL rewards approaches the ground-truth model and beats standard IRL.

## 7 CONCLUSION

**Discussion** In this work, we introduced Failure-Aware Inverse Reinforcement Learning (FA-IRL), a method that recovers the latent reward functions of LLMs by treating ambiguous or misclassified preference pairs as high-value signals rather than noise. We demonstrated both theoretically and empirically that this approach mitigates the core problem of non-identifiability in IRL, yielding more faithful and stable reward functions. Crucially, these rewards proved operationally superior, enabling downstream re-alignment that significantly reduced toxicity and closed the gap with ground-truth supervision, establishing FA-IRL as a robust tool for auditing and improving LLM alignment.

**Limitations** Our approach is not without limitations. The identification of failures currently relies on margin-based uncertainty, which may misinterpret preference ambiguity as error, or on supervised labels, which can be noisy or biased. Furthermore, our current taxonomy of failures is limited; it could be expanded to more explicitly account for complex behaviors like reward hacking, misgeneralization, or other systematic model misalignments. Finally, the fidelity of the recovered reward is fundamentally constrained by the preference data itself; our dataset was limited in scale (20,000 sampes) and intentionally curated for detoxification, meaning the resulting reward function is specialized to that task and may not capture the model's broader preferences that it has learnt. A truly holistic understanding of the learned reward landscape would necessitate a much larger and more diverse set of prompts and completions.

**Future Work** A necessary next step is to scale the FA-IRL framework to state-of-the-art language models (>70B parameters) and larger datasets. This will allow us to investigate its utility for more complex alignment tasks, such as reducing factual hallucination and ensuring faithfulness to source material. Future work can then be further broken into down into three key directions. First, we plan to develop more sophisticated failure detection methods with better uncertainty quantification. Second, we will investigate alternative supervision signals, such as task priors and consistency checks, to reduce reliance on external labels. Finally, we aim to extend FA-IRL to handle multi-objective alignment, allowing it to manage trade-offs between different types of failures and tasks simultaneously.

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

## A PRELIMINARIES: INVERSE REINFORCEMENT LEARNING APPROACHES

**Maximum Margin IRL.** Maximum Margin IRL is based on the intuition that the expert's policy should outperform all alternatives under the true reward function, with a defined margin. Assuming a linear reward model $R_\theta(s) = \theta^\top h(s)$, where $h(s)$ is a feature representation of the state and $\theta$ is a parameter vector, the expected feature counts for a policy $\pi$ are given by:

$$\mu(\pi) = \mathbb{E}\left[\sum_{t=0}^{\infty} \gamma^t h(s_t) \mid \pi\right] \quad (5)$$

The method imposes the following margin-based constraint to ensure separation between the expert policy $\pi_E$ and any suboptimal policy $\pi_0$:

$$\theta^\top \mu(\pi_E) \geq \theta^\top \mu(\pi_0) + 1, \quad \forall \pi_0 \neq \pi_E \quad (6)$$

This constraint, inspired by large-margin classifiers such as SVMs, enforces that the expert's behavior is not only optimal under $\theta$ but also sufficiently distinct from alternatives. The constant margin (e.g., 1) is arbitrary and can be rescaled alongside $\theta$ without affecting the solution. The learning objective is to find a parameter vector $\theta$ that satisfies the constraint in Eqn. 6 for all policies while maximizing the margin. While this is the general, multi-step formulation, in our single-step contextual bandit setting, it simplifies to enforcing a margin directly on the final outputs $o^+$ and $o^-$.

**Maximum Entropy IRL.** Maximum Entropy IRL is based on the principle of maximum entropy, which suggests that among all reward functions consistent with the expert demonstrations, we should choose the one that makes the fewest additional assumptions. Assuming a linear reward model $R_\theta(s) = \theta^\top h(s)$, where $h(s)$ is a feature representation of the state and $\theta$ is a parameter vector, the expected feature counts for a policy $\pi$ are given by:

$$\mu(\pi) = \mathbb{E}\left[\sum_{t=0}^{\infty} \gamma^t h(s_t) \mid \pi\right] \quad (7)$$

The method seeks to find a distribution over policies that matches the expert's feature expectations while maximizing entropy. Under the exponential family assumption, the optimal policy distribution takes the form:

$$P(\pi \mid \theta) = \frac{1}{Z(\theta)} \exp\left(\theta^\top \mu(\pi)\right) \quad (8)$$

where $Z(\theta) = \sum_\pi \exp(\theta^\top \mu(\pi))$ is the partition function. The learning objective maximizes the likelihood of the expert demonstrations while incorporating a maximum entropy prior:

$$\max_\theta \log P(\pi_E \mid \theta) - \lambda\|\theta\|_2^2 \quad (9)$$

This formulation naturally handles the ambiguity inherent in inverse reinforcement learning by preferring simpler explanations (higher entropy) when multiple reward functions could explain the data. The regularization term $\lambda\|\theta\|_2^2$ prevents overfitting and ensures numerical stability.

## B THEORETICAL GUARANTEES OF FAILURE-AWARE IRL

**Theorem 1 (Failures shrink the feasible reward set.)** *Consider FA-IRL with either the max-margin or max-entropy objective. In the max-margin case, if the margin on failure pairs is tightened ($M_{\text{fail}} > M$), then the feasible set of reward parameters under FA-IRL, $\mathcal{F}_{\text{FA}}$, is contained in the feasible set under standard IRL, $\mathcal{F}$:*

$$\mathcal{F}_{\text{FA}} \subseteq \mathcal{F},$$

*with strict inclusion whenever at least one nontrivial failure exists. In the max-entropy case, if failure pairs are up-weighted ($w_{\text{fail}} > 1$) or sharpened with a lower temperature ($\tau_{\text{fail}} < \tau$), then for any loss threshold $\eta$, the FA-IRL sublevel set is contained in the standard IRL sublevel set:*

$$\mathcal{F}_{\text{FA}}(\eta) \subseteq \mathcal{F}(\eta),$$

*again with strict inclusion whenever at least one failure term has positive loss. Thus, in both formulations, FA-IRL shrinks the feasible parameter space, thereby reducing the ambiguity of reward recovery.*

*Proof. Max-margin.* Write $x_i = \phi(o_i^+) - \phi(o_i^-)$ and $\Delta_i(\theta) = \theta^\top x_i$. For $i \in F$, FA-IRL replaces the base constraint $\Delta_i(\theta) \geq M$ by the stricter $\Delta_i(\theta) \geq M_{\text{fail}}$ with $M_{\text{fail}} > M$; for $i \notin F$ the constraint is unchanged. Hence every $\theta$ feasible for FA-IRL is feasible for the base problem, i.e., $\mathcal{F}_{\text{FA}} \subseteq \mathcal{F}$. Each margin constraint requires the reward parameter $\theta$ to lie on the side of a linear boundary where the preferred output scores higher than the non-preferred one. Increasing the required margin shifts this boundary, reducing the set of parameters that remain feasible. Thus, FA-IRL always yields a smaller feasible region whenever failures are present. Strict shrinkage occurs in any direction influenced by at least one failure pair (i.e., where some $x_i$, $i \in F$, has nonzero projection).

*Max-entropy.* Let

$$\ell_\tau(\Delta) = \log\left(1 + e^{-\Delta/\tau}\right)$$

denote the max entropy on a margin $\Delta$ with temperature $\tau$. The total base loss is

$$L_{\text{base}}(\theta) = \sum_i \ell_\tau(\Delta_i(\theta)).$$

In FA-IRL, failure pairs $i \in F$ are penalized more strongly, either by *up-weighting* or by *sharpening*:

$$L_{\text{FA}}(\theta) = \sum_{i \notin F} \ell_\tau(\Delta_i(\theta)) + \sum_{i \in F}\left[w_{\text{fail}}\ell_\tau(\Delta_i(\theta)) \ \text{ or } \ \ell_{\tau_{\text{fail}}}(\Delta_i(\theta))\right].$$

**Up-weighting case.** Here failures are multiplied by $w_{\text{fail}} > 1$. The difference from the base loss is

$$L_{\text{FA}}(\theta) - L_{\text{base}}(\theta) = (w_{\text{fail}} - 1)\sum_{i \in F} \ell_\tau(\Delta_i(\theta)) \ \geq 0.$$

Thus the FA-IRL loss is never smaller, and strictly larger whenever a failure has nonzero loss.

**Sharpening case.** Here failures use a smaller temperature $\tau_{\text{fail}} < \tau$. Since $\ell_\tau(\Delta)$ decreases as $\tau$ increases, it follows that

$$\ell_{\tau_{\text{fail}}}(\Delta) \ \geq \ \ell_\tau(\Delta)$$

for any finite $\Delta$, with strict inequality unless $\ell_\tau(\Delta) = 0$.

In both cases we obtain

$$L_{\text{FA}}(\theta) \ \geq \ L_{\text{base}}(\theta) \quad \text{for all } \theta,$$

with strict inequality whenever at least one failure pair contributes positive loss.

This pointwise dominance implies that the FA-IRL sublevel set

$$\mathcal{F}_{\text{FA}}(\eta) = \{\theta : L_{\text{FA}}(\theta) \leq \eta\}$$

is always contained in the base sublevel set

$$\mathcal{F}(\eta) = \{\theta : L_{\text{base}}(\theta) \leq \eta\}.$$

Hence FA-IRL reduces the feasible parameter space, and does so strictly whenever nontrivial failures exist. $\qquad\square$

**Corollary 1 (Reduction of non-identifiability.)** *Since FA-IRL produces strictly smaller feasible sets, any ambiguity measure that is monotonic under set inclusion (e.g., diameter, volume, directional width) cannot increase, and is strictly reduced whenever failures occur. Hence FA-IRL mitigates the non-identifiability of reward functions in IRL.*

*Proof.* The theorem shows FA-IRL yields a parameter set contained in the base set: $\mathcal{F}_{\text{FA}} \subseteq \mathcal{F}$ (max-margin) or $\mathcal{F}_{\text{FA}}(\eta) \subseteq \mathcal{F}(\eta)$ (max-entropy). Any ambiguity measure $A(\cdot)$ that is *monotonic under set inclusion* (i.e., $\mathcal{S}_1 \subseteq \mathcal{S}_2 \Rightarrow A(\mathcal{S}_1) \leq A(\mathcal{S}_2)$), such as diameter, directional width, or volume, therefore cannot increase under FA-IRL. When failures are nontrivial (some failure constraint tightens a supported face in max-margin, or some failure loss is positive in max-entropy), the inclusion is strict and the ambiguity measure strictly decreases, yielding reduced non-identifiability. $\qquad\square$

## C  EXPERIMENTAL SETUP DETAILS

**Evaluation Metrics.**   We evaluate standard IRL and FA-IRL along four complementary dimensions. First, *classification performance* is assessed by treating the recovered reward models as binary classifiers on preference pairs, reporting Accuracy, F1 score, and ROC-AUC. These metrics measure whether the reward function captures surface-level preference signals.

Second, we evaluate *reward fidelity* to assess how closely a recovered reward reflects the intended alignment objective. Scale and Translation Adjusted Reward Correlation (STARC) measures the distance between recovered reward scores $\hat{r}$ and ground-truth scores $r_{\mathrm{gt}}$ after canonicalization to remove scale and shift, with lower values indicating more faithful recovery and directly addressing IRL's non-identifiability:

$$\text{STARC}(\hat{r}, r_{\mathrm{gt}}) = \min_{a > 0,\, b \in \mathbb{R}} \ \frac{1}{N} \sum_{i=1}^{N} \big(\hat{r}(o_i) - (a \cdot r_{\mathrm{gt}}(o_i) + b)\big)^2, \tag{10}$$

where $a > 0$ rescales the ground truth reward, $b \in \mathbb{R}$ shifts the ground truth reward. The minimization finds the best affine transform to align scales.

Third, we conduct a *failure slice analysis*, reporting classification and fidelity metrics restricted to misclassified and near-tie pairs. This isolates whether FA-IRL improves specifically where standard IRL methods struggle.

Finally, we assess *downstream utility (Re-RLHF)* by fine-tuning base models with rewards extracted by each method and measuring toxicity rates and refusal correctness on held-out prompts. Improvements here demonstrate that FA-IRL rewards are not only more interpretable but also more effective for re-alignment.

## D  DATASET PAIR–MIX ANALYSIS

**Dataset composition materially affects the recovered reward: when informative T→NT pairs are scarce, all methods degrade and FA-IRL can underperform; as the share of learnable pairs increases, FA-IRL improves fastest and attains the lowest STARC.**

**Purpose.**  Quantify how the preference dataset's composition influences reward identifiability, and test whether FA-IRL benefits disproportionately from *learnable* comparisons (toxic→non-toxic, T→NT).

**Design.**  We fix the total number of pairs and sweep the *pair mix*—the fraction of T→NT pairs (lower values imply mostly NT→NT). For each mix we train all methods and report Training STARC (lower is better).

**Result.**  STARC decreases as the T→NT fraction increases, indicating better reward fidelity with more informative comparisons. At very low T→NT fractions (few/none), FA-IRL is comparable to or slightly worse than baselines, reflecting limited learnable signal; once the dataset includes a moderate share of T→NT pairs, FA-IRL improves most rapidly and achieves the best STARC.

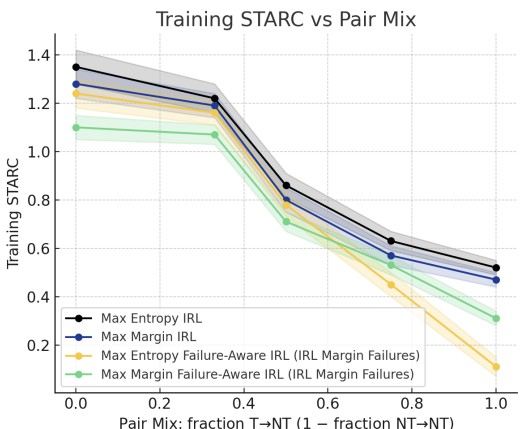

Figure 4: **Pair–mix sensitivity of reward fidelity.** Training STARC (lower is better) vs. fraction of T→NT pairs. Line shows the mean over 5 independent seeds; shaded band is the min–max range. Scarce T→NT pairs degrade all methods and can disadvantage FA-IRL; as learnable pairs increase, STARC drops overall, with FA-IRL improving fastest.

**Key takeaway: Curating a sufficient fraction of T→NT pairs is critical; FA-IRL leverages these informative comparisons best once they are present in moderate quantity.**

## E    MODEL SIZE ANALYSIS

**Purpose.** Evaluate how base model scale affects the *extractability* and stability of the recovered reward.

**Setup.** We run FA-IRL across four sizes (153M, 270M, 360M, 410M). For each size, we aggregate over five seeds and report mean *Accuracy* (higher is better) and *STARC* (lower is better).

**Result.** Mean performance strengthens from 153M to 360M (higher Accuracy, lower STARC, tighter intervals), followed by a modest drop at 410M. Given the narrow size range and the observed non-monotonicity at the top end, we interpret these results as *indicative* gains up to mid scale rather than evidence of an unbounded scaling law.

**Limitations.** Results are task and data specific and cover a limited parameter range; claims about larger models require broader evaluation.

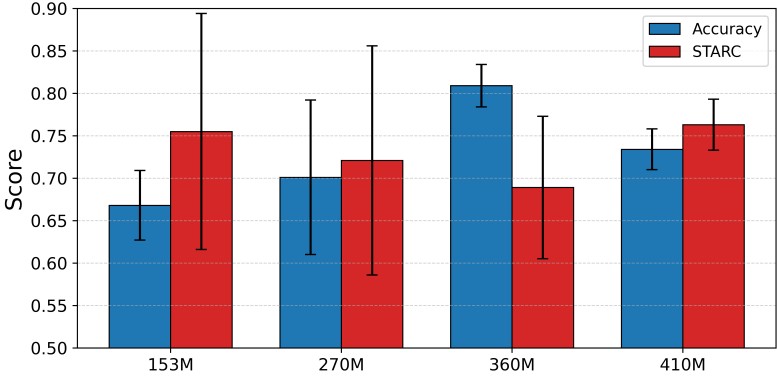

Figure 5: **Exploratory scaling (153M–410M, 5 seeds).** Mean *Accuracy* (blue) and *STARC* (red) with 95% CIs. Performance improves up to 360M, then slightly declines at 410M; we therefore refrain from claiming monotonic scaling beyond this range.

**Key takeaway: Gains accrue up to mid-scale (360M) in our setting; the 410M dip cautions against overclaiming scaling without larger models and broader tasks.**

