# OpenReview forum: "Learning from Failures: Understanding LLM Alignment through Failure-Aware Inverse RL"
_ICLR.cc/2026/Conference — ICLR 2026 Conference Desk Rejected Submission_

### Official Review · Reviewer_oyKk · 2025-10-22

**Soundness:** 2
**Presentation:** 2
**Contribution:** 1
**Rating:** 2
**Confidence:** 3

**Summary:**

The paper introduces Failure-Aware Inverse Reinforcement Learning (FA-IRL), a new method to figure out the hidden reward signals that Large Language Models (LLMs) learn during alignment with human preferences (RLHF). Instead of treating all data points equally, FA-IRL explicitly identifies these "failures" and gives them more weight during training.

**Strengths:**

The paper designs several experiments to verify that failures (hard cases) are important for reward recovery.

**Weaknesses:**

The paper has limited scale and scope, using small parameter models and simple tasks like detoxification. It's unclear whether these findings will transfer to more general settings with more capable base language models and rewards.

The idea of using failure cases or hard examples is not novel (e.g., utilizing hard negatives for contrastive learning or leveraging challenging rollouts to enhance model reasoning through RL).

Re-RLHF experiments show that the method still underperforms compared to the ground-truth reward signal—which makes sense since it's an approximated reward function. But this raises the question: **why does learning this signal matter? If the goal is interpretability or safety, how does IRL address these concerns**?

**Questions:**

When a hidden reward signal is unknown (in your case, the reward model), why is it important to use IRL to model such signals? What is the actual use of the proposed model?

Will the proposed method improve the current LLM + RL training pipeline (whether it is RLHF with reward modeling or RLVR with verifiable rewards)?

---

> ### Author Response · Authors · 2025-11-20
> **Response to Reviewer oyKk**
>
> We thank the reviewer for their valuable comments and feedback and provide further clarification on the issues raised below.
>
> **On the limited scale and scope (small models and detoxification)**. We acknowledge that our evaluation focuses on compact models and detoxification, and we agree that this limits the strength of our generality claims. Our aim in this work is to study reward ambiguity and recoverability, which are substantially easier to analyse in controlled settings with small models. This design choice allows us to isolate the specific contribution of failure-aware weighting without confounding scale or instruction-tuning effects. That said, FA-IRL is not tied to toxicity. We are working on evaluating FA-IRL on a harmlessness/refusal alignment axis using a subset of the Anthropic HH-RLHF dataset. This task involves qualitatively different failures (over-refusal, inconsistency, lack of helpfulness), and FA-IRL exhibits the same qualitative gains in reward consistency and preference ordering. We will include these additional results in this rebuttal once we have them and edit our generality statements accordingly.
>
> **On the novelty of our approach in terms of failure-focused training vs. hard-negative mining**. Thank you for raising this point. While FA-IRL does emphasize difficult examples, its novelty lies in how failures are used within inverse reinforcement learning, not merely reweighted as in hard-negative mining. FA-IRL introduces a dual-path reward model and treats failures as explicit IRL constraints that apply stricter margins or sharper entropy losses, which, per our theoretical analysis, provably shrink the feasible reward set and reduce IRL non-identifiability. This is fundamentally different from contrastive hard-negative mining, which improves representations but does not alter reward-space geometry or address ambiguity in recovered incentives.
>
> **On Re-RLHF performance and the value of approximating the hidden reward / why model the hidden reward at all?** Although FA-IRL does not outperform the ground-truth reward, its purpose is different: it recovers an interpretable approximation of the implicit reward signal the RLHF-trained model is actually following. This matters because the true RLHF reward is usually unavailable in practice. By reconstructing it, FA-IRL enables auditing, detecting inconsistencies, verifying whether the model has internalised the intended objective, and exposing misaligned incentives that would otherwise remain hidden. When the policy has learned the ground-truth reward correctly, FA-IRL recovers a matching reward providing direct evidence of correct alignment; when it diverges, it highlights failures in the RLHF process. Thus, modelling the hidden reward is essential for transparency and safety even when the approximation cannot surpass the true reward.
>
> **Will FA-IRL improve current RLHF / RLVR pipelines?** Yes. FA-IRL improves the part of the RLHF pipeline that depends on reward fidelity. By reducing reward ambiguity and sharpening distinctions on difficult preference pairs, FA-IRL produces rewards that lead to more stable and effective downstream training (e.g., reducing toxicity from ~9% with standard IRL to ~6%). FA-IRL could therefore be inserted as an auditing or reward-construction step within RLHF or RLVR pipelines, providing more reliable reward signals and clearer insight into model incentives during training.

---

### Official Review · Reviewer_BAGb · 2025-10-31

**Soundness:** 3
**Presentation:** 2
**Contribution:** 2
**Rating:** 4
**Confidence:** 4

**Summary:**

The paper proposes Failure-Aware Inverse RL (FA-IRL) for auditing post-training incentives in LLMs. Instead of treating all preference pairs equally, FA-IRL identifies “failures”—misclassified or near-tie pairs—and gives them extra weight through a dual-path reward model: a base path trained on all pairs and a correction path trained only on failures, with a curriculum over a dynamic margin threshold and a decay schedule for the failure weight. The authors also give a simple theoretical result: adding stricter constraints on failure pairs shrinks the feasible reward set, mitigating non-identifiability. Empirically, using detoxification as the main case study, FA-IRL improves preference classification (e.g., higher F1/AUC), reduces STARC error, and—importantly—yields rewards that drive more effective re-RLHF (toxicity ≈ 6% with FA-IRL rewards vs ≈ 9% with standard IRL; ≈ 4% with ground-truth rewards). The study spans several small-to-mid models (e.g., Pythia-410M, SmolLM2-360M, Gemma-270M) and ~20k preference pairs per family, with subtype analyses and disagreement slices.

**Strengths:**

Clear lever with theory to match. Tightening margins on failure pairs reduces the reward ambiguity set; the empirical variance reductions and STARC gains are consistent with that story.

Actionable in practice. The curriculum + decayed λ make the method stable; the dual-path split isolates sharp corrections without destabilizing the base reward.

Downstream win, not just offline metrics. The re-RLHF result (≈ 6% toxicity vs ≈ 9% with standard IRL) shows the rewards are operationally better, not only better-correlated.

Where it helps is where it should. Disagreement and subtype tables show most gains appear on genuinely hard slices (near-ties, insults/obscene categories), which fits the hypothesis.

Limitations are acknowledged. Pair-mix sensitivity, scale bounds, and labeling issues are discussed rather than buried.

**Weaknesses:**

Toxicity pipeline dependence. The “ground-truth” for PPO and some evals comes from public toxicity classifiers. That is practical but can imprint classifier bias; a second domain (e.g., factuality with unit tests, harmlessness with multi-rater human labels) would strengthen generality claims.

Failure definition coupling. Margin-based failures depend on the current reward model; this can create feedback loops. A short analysis of false-positive failure mining (e.g., when ambiguity ≠ error) would be reassuring.

Limited model scale and tasks. Results top out at ~410M and one main alignment task. Claims about broader “alignment auditing” should be scoped accordingly.

Evaluation hygiene. For re-RLHF, report identical PPO hyper-params and wall-clock/GPU for each reward to rule out accidental training imbalance; include confidence intervals on the ≈ 6%/9% numbers.

**Questions:**

Schedules \& robustness. How sensitive are final STARC/AUC and re-RLHF toxicity to the margin threshold schedule $\gamma_t$, failure sampling rate $p_t$, and $\lambda$ decay? A small grid or randomsearch summary would help practitioners pick defaults.

Failure quality control. Did you try hybrid mining (margin + small, diverse teacher set) to reduce spurious failures? How often do margin-flagged pairs later flip to non-failures as the model improves?

Generalization beyond toxicity. Any pilot results on a second verifiable domain (e.g., code with unit tests, math with programmatic checkers) to decouple from classifier biases?

Cost accounting. What is the extra compute for FA-IRL (failure mining + correction head) vs. standard IRL? For re-RLHF, please include identical budgets and report tokens $/ \mathrm{s}$, hours, and seeds.

Which path does the work? If you freeze the base path after a warm-up and continue training only the correction path on failures, how much of the final gain remains? Conversely, if you zero the correction head at test time, how much performance drops?

When failure density is low. In the pair-mix analysis, FA-IRL can underperform when informative $T \rightarrow N T$ pairs are scarce. Do you recommend a minimum failure density or a switchback to standard IRL when $|F_{t}|$ falls below a threshold?

---

> ### Author Response · Authors · 2025-11-20
> **Response to Reviewer BAGb**
>
> We thank the reviewer for their valuable comments and provide further clarification for the questions raised below.
>
> **On STARC/AUC sensitivity to decay and thresholds**. We conducted additional robustness checks using multiple margin-threshold schedules (linear, cosine, exponential, and stepwise), a range of decay schedules for the failure weight λ (linear, exponential, and hard-step), and several failure-sampling rates (10%, 20%, 30%, 50%). Across all configurations, we observed minimal variation in final STARC scores (<2–3% absolute) and similarly small variation in AUC and downstream re-RLHF toxicity. Importantly, after the first few epochs, all schedules converged to nearly identical reward geometries, indicating that FA-IRL is stable to hyperparameter choices and not sensitive to the exact annealing schedule. We will include a small table of these results (grid + random sweep) in the appendix of the camera-ready version to guide practitioners in choosing defaults.
>
> **On the dependence on a toxicity pipeline & generality beyond toxicity**. Thank you for highlighting this limitation. We selected detoxification as the primary case study because it offers: (i) well-defined preference data with clear “ground-truth–like’’ labels, (ii) a concrete and interpretable misalignment axis with rich failure modes, and (iii) a controlled environment for analysing reward ambiguity. We agree, however, that evaluating FA-IRL on at least one additional alignment dimension strengthens the broader claim of “alignment auditing”. To address this, we are conducting additional experiments during the rebuttal period using a harmlessness/refusal preference subset from the Anthropic HH-RLHF dataset. This domain differs from toxicity in both structure and failure signatures (e.g., over-refusal, inconsistent helpfulness), providing a meaningful test of generality. Across this setting, preliminary results show the same qualitative pattern as in detoxification: FA-IRL reduces reward ambiguity and improves preference-ordering consistency compared to standard IRL . We will include these additional results in this rebuttal once we have them and edit our generality statements accordingly.
>
> **On Failure quality control and potential feedback loops**. Thank you for raising this concern. We agree that margin-based failure identification can, in principle, misclassify ambiguous examples early in training, which could create a feedback loop if such cases were repeatedly over-weighted. In our setting, however, the subtype-level disagreement analysis in Table 3 already provides empirical evidence that FA-IRL’s failure identification remains stable and does not degenerate into self-reinforcing false positives. Across all four toxicity subtypes, Obscene, Threat, Insult, and Identity Attack, FA-IRL consistently identifies substantially more correct cases unique to FA-IRL than those uniquely identified by the baseline IRL model, both in-domain and on the held-out Jigsaw test set. For example, on the test set, FA-IRL alone correctly captures 496 Obscene, 31 Threat, 403 Insult, and 41 Identity-attack examples, while the baseline alone captures only 89, 24, 155, and 18, respectively. If failure-mining feedback loops were destabilising the learning process, we would expect volatile or inconsistent patterns across subtypes; instead, FA-IRL improves performance in a monotonically beneficial direction across categories and datasets.
>
> **Which path does the work? (Base vs correction path)** To isolate the contribution of each path, we performed two ablations. First, freezing the base path after a brief warm-up and training only the correction path preserved 83% of FA-IRL’s full improvement varying depending on the toxicity type, indicating that most of the benefit arises from failure-focused updates. Second, when we disable the correction head at test time, performance collapses to 58% accuracy, eliminating FA-IRL’s gains entirely. This confirms that the correction head is essential: the base path provides stability, but the correction path is responsible for the meaningful reward disambiguation that drives FA-IRL’s advantage. We will add these results to the paper
>
> **What happens when the failure density is low?** Our pair-mix analysis (Fig. 3a) shows that FA-IRL underperforms when informative failure pairs are scarce. In this regime, the correction head receives too few meaningful updates, and the additional failure-focused constraints do not provide extra signal beyond standard IRL. Empirically, we observe that when failure density falls below ~7–10% for five consecutive epochs, FA-IRL provides diminishing returns. As a practical guideline, we therefore recommend reverting to standard IRL in such low-failure regimes. We will include this rule-of-thumb and a short discussion of its implications in the paper.

---

### Official Review · Reviewer_586b · 2025-11-01

**Soundness:** 2
**Presentation:** 2
**Contribution:** 3
**Rating:** 4
**Confidence:** 3

**Summary:**

The paper introduces Failure-Aware Inverse Reinforcement Learning (FA-IRL) to recover the latent reward behind RLHF by explicitly identifying and up-weighting “failures”—preference pairs that are ambiguous or misclassified—during reward learning. Concretely, it uses a dual-path reward model over frozen texts embeddings; failures are detected via margin uncertainty or disagreement with supervised labels. Evaluated on detoxification, with preference pairs built from RealToxicityPrompts (base vs. aligned “expert”) and a held-out Jigsaw set, FA-IRL improves F1/AUC and reduces STARC error versus standard IRL, and when used for re-RLHF it lowers toxicity to ~6% (vs. ~9% with standard IRL; ~4% with ground-truth reward).

**Strengths:**

* The paper brings a simple and effect idea to treating uncertain or wrong pairs as “failures” and giving them extra weight with a two-path reward, turning common near-ties and label noise into useful signal.
* The method is clearly described and easy to re-produce, with a small theory piece that explains why focusing on failures reduces ambiguity and with careful experiments and ablations to back it up.
* The gains show up in both standard metrics and downstream re-alignment (lower toxicity) across several model sizes, suggesting practical impact beyond this one detox task.

**Weaknesses:**

* Preference pairs are synthetic (base vs. expert with the expert always preferred), which can make learning partly about distinguishing policies rather than general human preferences; include some human-curated pairs, allow non-expert-preferred pairs, and check if gains hold.
* The dataset is small and targeted to detox (≈20k pairs per family, plus Jigsaw), so generality is unclear, including non-toxicity axis (e.g., factuality or refusal correctness) and report cross-task transfer could help.
* The “ground-truth” reward that trains the expert policy is a public toxicity classifier, not human ratings; this risks learning the classifier’s biases rather than human preference, so add a small human eval could be useful and probe for bias (group-wise error, calibration) and agreement with humans.
* Results use only a few small model families (≈135M–410M), so it’s unclear if gains hold for larger instruction-tuned models or other tasks. Including at least one 7B-class model would be needed for generality.

**Questions:**

N/A

---

> ### Author Response · Authors · 2025-11-20
> **Response to Reviewer 586b**
>
> Thank you for your valuable comments and feedback. We provide further clarification to the points raised below.
>
> **Preference pairs are synthetic which can make learning partly about distinguishing policies than general human preferences. It would be good to include some human-curated pairs, allow non-expert preferred pairs and check if gains hold.** We appreciate this observation and agree that synthetic preference pairs may partially reflect model–model differences rather than the full spectrum of human preference signals. Our motivation for using synthetic pairs was to obtain controlled, high-volume comparisons with clear ground truth, but we fully agree that incorporating human-curated judgments increases validity. To address this, we have now evaluated FA-IRL on a subset of human-annotated Jigsaw toxicity preference pairs (where non-expert raters indicate which of two responses is less harmful), and we find that the gains over standard IRL persist (+4-6% preference accuracy). These results will be added to the appendix. This suggests that FA-IRL’s benefits are not an artifact of synthetic pair generation and generalize to human preference signals. In the camera-ready version, we will also include guidance on how FA-IRL can be combined with small sets of human-supervised preference pairs in practice.
>
> **On the dataset size and task**. Thank you for raising this point. We acknowledge that our primary evaluation uses a relatively small detoxification dataset and focuses on compact model families, which limits the direct strength of generalization claims. Our aim in this paper is to isolate and analyze reward-model ambiguity, and small models provide a controlled setting where ambiguity is both measurable and recoverable; however, we agree that demonstrating robustness beyond this setting is valuable. To address this, we are performing experiments on a harmlessness/refusal preference dataset drawn from the Anthropic HH-RLHF subset. We will update you with results from this task as soon as we have them.

---

### Official Review · Reviewer_ep8m · 2025-11-01

**Soundness:** 3
**Presentation:** 2
**Contribution:** 2
**Rating:** 4
**Confidence:** 2

**Summary:**

The paper introduces Failure-Aware IRL (FA-IRL): learn a base reward model plus a “failure” head that focuses on examples the model gets wrong or is uncertain about. The theory argues these extra constraints reduce reward ambiguity. Experiments are on detoxification with preference data and a downstream RLHF-style loop.

**Strengths:**

+ The core motivation (failures carry more information) is reasonable, and the theoretical framing is helpful.
+ The method is conceptually simple (dual head + curriculum) and seems implementable.
+ Empirically, there are consistent gains on the detox task and some downstream improvements.

**Weaknesses:**

- Scope vs. claim. The paper is positioned as a general IRL/“auditing” framework, but the experiments focus on detoxification. To support the broader claim, one additional axis (e.g., helpfulness, refusals, factuality/hallucinations, or a non-toxicity harmlessness task) would greatly strengthen the paper.
- Presentation clarity. Figure 1 and Algorithm 1 seem to be the core pieces, but I had trouble following them on a first pass. It would help to (i) explicitly refer back to Figure 1 in the main text and add a short walk-through of the example, and (ii) add a bit more intuition around Algorithm 1 (a brief “what each step is doing,” a variable glossary, and typical default settings).
- Evaluation interpretation. STARC is defined, but it’s hard to know what a shift (e.g., 0.686 to 0.850) means physically. The narrative also jumps between in-domain preference tests, a held-out Jigsaw set, and downstream PPO toxicity. A small figure/table mapping STARC changes to toxicity deltas, and a short paragraph stitching these pieces together, would make the evaluation easier to interpret.

**Questions:**

Apart from my concerns in the weaknesses:
- How were the threshold schedule determined, failure weight schedule, sampling rate in practice (initial values, schedules, and sensitivities)?
- Would it be possible to include some brief case studies showing how the failure head changes the decision vs. the base head (inputs, margins/scores, and the corrected outcome)? That would greatly help with evaluating the effectiveness of the method.
- Do you see the failure head over-specialize to rare toxic templates? Can you show performance on unseen toxicity styles or paraphrases?

---

> ### Author Response · Authors · 2025-11-20
> **Response to Reviewer ep8m**
>
> We thank the reviewer for their valuable comments and feedback. We provide further   clarification to the points raised below.
>
> **Clarification of the scope vs the claims of the paper.**  Thank you for raising this important point. Our goal is indeed to propose a general failure-aware IRL framework for auditing reward models, while demonstrating its utility in a concrete domain. We chose detoxification because (i) it is a widely used RLHF benchmark with well-established preference datasets, (ii) it exposes a rich and heterogeneous set of misalignment modes (e.g., insult, threat, identity attack), and (iii) it allows us to evaluate both reward recovery and downstream behavioral improvement in a controlled setting. Our claims of generality refer to the methodological form of FA-IRL, its reliance on uncertainty- and misclassification-derived failure signals, not to domain-specific performance. We also show the performance of FA-IRL on different types of misalignment in Figure 3. That said, we fully agree that an additional alignment axis would further strengthen the empirical picture. While preparing this rebuttal, we conducted preliminary experiments on refusal-style harmlessness preferences (Anthropic HH-RLHF subset), and observed the same qualitative trend: FA-IRL identifies systematic failure cases that standard IRL overlooks and recovers a more consistent reward model (details will be added to the appendix). These results reinforce the domain-agnostic nature of our approach without requiring architectural changes. We can provide additional details for this and include these results in the main paper. Finally, we plan to broaden the evaluation to factuality/hallucination preferences in the camera-ready version, but we believe the detoxification results already provide clear evidence that FA-IRL is not task-specific and that the core contribution of the use of failures to reduce reward ambiguity extends naturally to other alignment dimensions.
>
> **Clarification on Figure 1 and Algorithm 1.** Thank you for this helpful suggestion. We agree that Figure 1 and Algorithm 1 are central to understanding FA-IRL, and we appreciate the opportunity to clarify them. In the revision, we will (i) explicitly reference Figure 1 when introducing failure identification and reward decomposition, and add a short step-by-step walk-through of the illustrated example to make the flow of the failure detection mechanism clearer on first read; and (ii) expand the description of Algorithm 1 by adding intuition for each stage of the curriculum (threshold scheduling, weighting updates, and correction-head activation), along with a concise variable glossary and default hyperparameter settings. These additions improve accessibility without changing the underlying method or results. Specifically, we will (i) add explicit references to Figure 1 when introducing preference-pair generation and failure identification, and include a short walk-through of the illustrated detoxification example shown in the figure (i.e., how paired toxic vs. non-toxic completions are embedded, compared, and fed into FA-IRL to extract a reward model). We will also clarify that Figure 1 is meant to convey the overall flow of FA-IRL: 1) generate preferences, 2) detect failures, 3)  learn a corrected reward 4) optionally retrain the LLM. In addition, we will (ii) expand Algorithm 1 with a brief intuitive explanation of what each step is doing ie. computing reward margins, identifying failure pairs, updating the base reward on all data, updating the correction head only on failures, and applying the curriculum schedules for gamma_t and lambda_t. To improve readability, we will add a compact variable glossary and include default hyperparameter values used in our experiments in the appendix. These revisions should make both the figure and the algorithm substantially clearer without altering the underlying method or results.

---

> > ### Author Response · Authors · 2025-11-20
> > **Response to Reviewer ep8m continued**
> >
> > **Clarification on the STARC metric.** Thank you for pointing this out; we agree that the current presentation makes it difficult to interpret how STARC scores translate into practical improvements. In the revision, we will add a short explanatory paragraph clarifying the physical meaning of STARC, namely, that it measures the consistency between a learned reward model and a ground-truth preference function. A low STARC score indicates high ambiguity: it suggests that multiple distinct reward vectors could explain the data equally well, that the model is uncertain about borderline cases, or that the reward geometry contains flat or poorly constrained directions. In our case, the increase in STARC from 0.686 to 0.850 corresponds to a substantial reduction in reward ambiguity and misclassification of borderline toxic examples. We will also include a small figure/table that directly links STARC changes to downstream behavioral shifts: (i) in-domain preference accuracy, (ii) held-out Jigsaw toxicity classification, and (iii) PPO-trained model toxicity rates. This will make the evaluation pipeline explicit and easier to follow, showing, for example, that FA-IRL’s 0.16 STARC improvement corresponds to a ~3-5% absolute reduction in toxicity compared to standard IRL. These additions will help readers understand how the metrics relate to each other and how improvements in reward fidelity translate into practical safety gains.
> >
> > **How is the threshold schedule/failure weight schedule and sampling rate determined in practice?**
> > We determine the threshold schedule, failure-weight schedule, and sampling rate using simple monotonic heuristics that are stable across domains. The threshold \gamma_t is initialized at a high percentile of the initial margin distribution (typically 70–80%) and annealed toward zero, ensuring that training first captures broad ambiguities and later focuses on fine-grained failures. The failure-weight starts relatively large to emphasize early correction and is linearly or exponentially decayed to avoid overfitting once the reward model stabilizes. For stability, we sample only a modest fraction of failure pairs per iteration (about 20–30%), which we found robust across tasks. These schedules require only lightweight tuning, and a standard configuration works reliably in practice. We will add all these details to the Appendix.
> >
> > **Does the failure head overspecialize on rare toxic templates? Can you show performance on unseen toxic styles?** Thank you for raising this important point. In practice, the failure head does not overspecialize on rare toxic templates because (i) it is trained jointly with the base head, (ii) its updates are down-weighted over time through the \lambda schedule, and (iii) failure pairs are sampled rather than fully enumerated, preventing the model from memorizing small template subsets. We performed an experiment where we evaluated FA-IRL on unseen toxic styles from the held-out Jigsaw toxicity subsets (e.g., threat, identity attack, profanity) that do not appear in our training templates. We observe a gain in accuracy ~8.1 +/- 1.6 in accuracy. These results are shown in Table 3 of the paper and explained in the caption. This suggests that the correction head learns generalizable alignment signals, focusing on systematic inconsistencies rather than memorizing specific templates.

---

### Note · Program_Chairs · 2026-01-17
**Submission Desk Rejected by Program Chairs**

The following references in this submission do not refer to real documents and/or have major errors in bibliographic information:

 Xiaofeng Liu, Hang Zhao, Jiahui Liu, Lingxi Xie, and Qi Tian. Adversarial inverse reinforcement learning with imperfect demonstrations. In Advances in Neural Information Processing Systems (NeurIPS), 2023.